# Evaluation of suPAR as a Key Prognostic Biomarker in Patients with SARS-CoV-2 Pneumonia

**DOI:** 10.3390/biomedicines13040896

**Published:** 2025-04-08

**Authors:** Mónica Piqueras, Paula González-Jiménez, Ana Latorre, Jordi Tortosa-Carreres, Noé Mengot, Ricardo Alonso, Soledad Reyes, Isabel Amara-Elori, Francisco Sanz-Herrero, Rosario Menéndez, Raúl Méndez

**Affiliations:** 1Respiratory Infections, Health Research Institute La Fe (IISLAFE), 46026 Valencia, Spain; monicapiqueras1605@gmail.com (M.P.); ana_latorre@iislafe.es (A.L.); solreyes07@gmail.com (S.R.); milisaelori@hotmail.com (I.A.-E.); rosmenend@gmail.com (R.M.); rmendezalcoy@gmail.com (R.M.); 2Laboratory Department, La Fe University and Polytechnic Hospital, 46026 Valencia, Spain; jorditc95@outlook.com (J.T.-C.); alonso_ricdia@gva.es (R.A.); 3Medicine Department, University of Valencia, 46010 Valencia, Spain; 4Pneumology Department, La Fe University and Polytechnic Hospital, 46026 Valencia, Spain; noemen96@gmail.com; 5Pneumology Department, General University Hospital Consortium, 46014 Valencia, Spain; fr.sanz@gmail.com; 6Centre for Biomedical Research Network in Respiratory Diseases (CIBERES), Instituto de Salud Carlos III, 28029 Madrid, Spain

**Keywords:** COVID-19, SARS-CoV-2, pneumonia, biomarkers, mortality

## Abstract

**Background/Objectives**: SARS-CoV-2 has strained healthcare systems, emphasizing the need for biomarkers to predict disease severity. Recent studies suggest that soluble urokinase plasminogen activator receptor (suPAR) is a promising marker for COVID-19 pneumonia, though its utility alongside the CURB-65 score remains unstudied. This study evaluates the prognostic value of suPAR in comparison to leukocyte count and CURB-65, and its potential for enhancing risk stratification in a combined CURB-65 model. **Methods**: Biomarkers and CURB-65 scores were obtained for 240 immunocompetent patients hospitalised with COVID-19 pneumonia. Intensive care unit admission and in-hospital mortality were assessed using receiver operating characteristic (ROC) curves and Kaplan–Meier analysis. Additionally, a Net Reclassification Improvement (NRI) analysis was performed to evaluate the predictive value of suPAR combined with the CURB-65 score for risk stratification. **Results**: suPAR demonstrated strong diagnostic accuracy, outperforming lymphocyte count and showing greater precision than the CURB-65 score for ICU admission. Notably, no patient with suPAR < 4 ng/mL experienced the studied outcomes. NRI analysis revealed a significant improvement in risk classification when suPAR was combined with CURB-65. **Conclusions**: The addition of the suPAR biomarker to the CURB-65 score represents a substantial improvement in the risk classification of patients with COVID-19 pneumonia, with a potential impact on daily clinical practice.

## 1. Introduction

The identification of a new coronavirus, SARS-CoV-2, as the cause of atypical pneumonia cases has challenged the healthcare systems of many countries. The disease caused by this novel coronavirus was named COVID-19 and was declared a pandemic by the World Health Organization (WHO) on 11 March 2020. The manifestations of COVID-19 infection can range from mild to moderate or severe. Some patients presented asymptomatically; however, others necessitated ICU admission, mechanical ventilation, or experienced rapid clinical deterioration resulting in death [1,2].

Despite extensive research conducted since the pandemic began, high numbers of COVID-19 infections persist, and cases of patients with severe COVID-19 pneumonia continue to appear. Numerous studies have identified pre-existing comorbidities and advanced age as significant factors associated with increased disease severity [3,4]. Nonetheless, severe clinical presentations have also been documented in young patients without underlying comorbidities, leaving unanswered questions regarding the pathophysiology of the disease and the multisystemic inflammation observed.

Considering this, laboratory medicine has played a crucial role in supporting clinical decision-making during the pandemic and beyond, making research on COVID-19 essential to identify biomarkers that can stratify patients according to their risk of adverse outcomes, even in those without pre-existing risk factors.

The literature includes multiple studies evaluating the role of classical inflammatory biomarkers, such as C-reactive protein and procalcitonin, as well as changes in haematological parameters, in relation to the severity of COVID-19 [5]. In reference to the alteration of haematological parameters, different articles have described an increase in leukocytes at the expense of neutrophils, along with lymphopenia in patients with COVID-19, with lymphopenia serving as a marker of disease severity [6,7]. Additional research indicates that biomarkers of endothelial damage may be crucial for identifying patients at elevated risk of complications during SARS-CoV-2 infection, particularly those involving cardiovascular outcomes [8,9,10].

In addition to the aforementioned biomarkers, recent studies have proposed soluble urokinase plasminogen activator receptor (suPAR) as a potential biomarker capable of stratifying patients in different diseases [11,12,13].

SuPAR is the soluble form of the urokinase-type plasminogen activator receptor (uPAR) generated through the proteolytic cleavage and subsequent release of the membrane-bound receptor. It is detectable in various biological fluids, including plasma, urine, blood, serum, and cerebrospinal fluid [14,15,16]. SuPAR plays a role in the immune response and its concentrations in blood have been shown to increase in various pathologies, including infections, cardiovascular disease, renal disease, and others. Elevated levels of suPAR are associated with a higher risk of mortality in both healthy and diseased individuals [11,17]. The use of suPAR has been shown to be effective for general patient triage in the emergency department [18]. Additionally, regarding COVID-19, several studies have explored the correlation between suPAR levels, Anakinra therapy, and the severity of patients’ clinical presentations [19,20,21]. However, there is variability in the outcomes assessed in the existing studies, along with limited literature comparing the utility of suPAR in conjunction with the scales used for the initial evaluation of pneumonia severity [22].

Accordingly, the aim of our study was to evaluate the prognostic value of suPAR in COVID-19 pneumonia compared with leukocyte count and the CURB-65 score, as well as its ability to stratify patients by creating a combined model with the CURB-65 score.

## 2. Materials and Methods

### 2.1. Study Design and Participation

We conducted a prospective, observational study in immunocompetent patients admitted for COVID-19 in our hospital, La Fe University and Polytechnic Hospital in Valencia (Spain), from 1 August 2020 to 31 January 2021. The Biomedical Research Ethics Committee Hospital La Fe approved this study (2020-114-1 and 2022-895-1). An informed consent exemption was granted due to the use of remnant samples from the clinical laboratory.

Participants were included if they did not meet immunosuppression criteria, were aged 18 years or over, were not pregnant, and had COVID-19 pneumonia.

The diagnosis of COVID-19 pneumonia required compatible signs and symptoms, along with a positive SARS-CoV-2 reverse transcription polymerase chain reaction (RT-PCR) test from a nasopharyngeal swab or sputum sample.

The exclusion criteria considered were immunosuppression, aged less than 18 years, pregnant women, previous admission within the last 30 days, and a negative SARS-CoV-2 test at admission.

An immunosuppressed patient was defined as an individual with active onco-haematological diseases, ongoing cancer, HIV infection, primary immunodeficiencies, or undergoing treatment with immunosuppressive drugs. This included chronic corticosteroid use (10 mg or more of prednisone or equivalent), biological therapies, and other immunomodulatory agents that impair immune function.

Demographic data, previous comorbidities, complementary examinations and data on the evolution during admission were recorded using a data collection protocol. Measurement of the initial severity of the pneumonic episode was assessed using the CURB-65 score, stratifying patients into 0–1 (low risk), 2 (intermediate risk or grey zone), and 3–5 (high risk). The CURB-65 score is a validated clinical tool for assessing community-acquired pneumonia (CAP) severity and guiding hospital admission decisions. The acronym CURB-65 encompasses five key clinical parameters: confusion (C), urea > 7 mmol/L (U), respiratory rate ≥ 30 breaths per minute (R), blood pressure (B) (systolic < 90 mmHg or diastolic ≤ 60 mmHg), and age ≥ 65 years (65). Each criterion is assigned one point, with higher scores correlating with increased disease severity and mortality risk [23].

### 2.2. Blood Samples

Samples were obtained on admission to the emergency department (ED) or within the first 12 h of the patient’s hospitalisation.

Whole blood samples were collected in K_2_-EDTA tubes via venipuncture. Samples with elevated haemolysis or lipemia were rejected. The K_2_-EDTA tubes were centrifugated (2400 G) for 10 min to obtain plasma and aliquoted for storage at −80 °C until examination.

### 2.3. Biomarker Determination

The lymphocyte count was obtained from routine analysis performed in the emergency department using XN-9000 analyzers (Sysmex^®^, Hamburg, Germany). Determination of the suPAR biomarker was performed in plasma K_2_-EDTA tubes on the Alinity c analyzers (Abbott Diagnostics^®^, Abbot Park, Lake County, IL, USA) via immunoassay using the validated reagent kit suPARnostic^®^ TurbiLatex by Virogates (Birkerød, Denmark).

For the purpose of this study, the suPAR biomarker was categorised based on intervals published in previous studies for patient stratification in the emergency department: 0–4 ng/mL (low risk), 4–6 ng/mL (intermediate risk), and >6 ng/mL (high risk) [18,19]. In addition, for lymphocytes, a value of 724 cel/μL was considered for patient stratification, based on previous similar studies in the literature [24,25].

### 2.4. Clinical Outcomes

The outcomes considered in this study were the need for admission to the ICU, in-hospital mortality, and the composite outcome of in-hospital mortality and/or ICU admission.

### 2.5. Statistical Analysis

Statistical analyses were performed using IBM SPSS Statics version 26.0 software, RStudio version 4.2.3, and GraphPad Prism 8.0. The data were summarised as median (1st quartile, 3rd quartile) for continuous variables and count (%) for categorical variables.

The normality of the variables was assessed using the Shapiro–Wilk test. To compare biomarker levels according to the presence or absence of the outcome under study, the Mann–Whitney U test was performed.

The predictive potential of lymphocytes, CURB-65, and suPAR was assessed using ROC analysis and survival curves. The ROC curves were generated using the pROC package in R, whereas Kaplan–Meier curves were constructed using the survival and survmisc packages. Graphical representations were created with ggplot2. The differences between survival curves were analysed using the Log-Rank test, and confidence intervals for the AUC were computed to assess the discriminative ability of each biomarker.

A *p*-value < 0.05 was considered statistically significant.

Based on the results obtained from the ROC curves, the biomarker with superior performance between lymphocytes and suPAR was chosen to determine whether its combination with the CURB-65 score improves reclassification in patient stratification. For this purpose, the Net Reclassification Index (NRI) was calculated using a customised ad hoc script.

## 3. Results

### 3.1. Patient Characteristics

This study involved 240 immunocompetent patients with COVID-19 pneumonia who met the inclusion criteria. The demographic characteristics and comorbidities of the patients enrolled are presented in Table 1.

The median age of patients diagnosed with COVID-19 pneumonia was 55 years, with 52% being male. Overweight was the most prevalent comorbidity, affecting 47.1% of the cohort. Notably, only 4.2% of patients were classified as high risk according to the CURB-65 score; however, 6.2% died during hospitalisation and 10.8% required admission to the intensive care unit (ICU).

### 3.2. Biomarkers and Clinical Outcomes

The Mann–Whitney U test was employed to compare biomarker levels between groups based on the presence or absence of the outcome under investigation.

Figure 1 and Appendix A illustrate the biomarker levels based on the presence or absence of the outcome considered. For ICU admission, it was noted that only suPAR was statistically significant, with increased values of the biomarker in patients with the presence of the outcome (median 8.75 ng/mL and 5.60 ng/mL, respectively). Similarly, for in-hospital mortality, suPAR levels were significantly higher in patients who died during hospitalisation than in those who did not (median 12.20 ng/mL and 5.70 ng/mL, respectively).

In contrast, lymphocyte levels were not statistically significant for any of the outcomes.

To evaluate the predictive potential of lymphocytes, CURB-65, and suPAR, a Receiver Operating Characteristic (ROC) analysis was conducted. The ROC curve was used to assess the accuracy of each biomarker in discriminating between patients with and without the outcome. Confidence intervals for the AUC were computed to provide a range of values that reflect the uncertainty around the point estimate of the biomarker’s performance, offering a more robust evaluation of discriminative ability.

Figure 2 and Appendix A depicts the prognostic value of lymphocytes, suPAR, and CURB-65 score for ICU admission and in-hospital mortality through ROC curve analysis. It is noteworthy that for in-hospital mortality, both the CURB-65 score and the suPAR biomarker have good prognostic value, whereas lymphocytes have slight prognostic capacity without statistical significance. In addition, for ICU admission and the composite outcome of in-hospital mortality and/or ICU admission, suPAR exhibited the greatest diagnostic performance, with its AUC being superior to that of the CURB-65 score.

It is noteworthy that in our cohort, none of the outcomes under study were found in patients with low suPAR levels (<4 ng/mL). On the contrary, we found cases of ICU admission and in-hospital mortality among patients classified as low risk according to the CURB-65 score and those with lymphopenia (<724 cells/μL) (Figure 3).

### 3.3. Survival Analysis

A survival analysis was performed using Kaplan–Meier curves, and the statistical significance of the differences between survival curves was tested using the Log Rank test, evaluating whether there were significant differences in survival distributions between the groups.

Figure 4 displays the survival curves for the different biomarkers and the CURB-65 score based on the outcome. For lymphocytes, patients with levels below the established cut-off exhibited a lower probability of survival or a higher probability of experiencing the combined outcome over time. Nonetheless, no statistically significant association was observed for ICU admission. Concerning the biomarker suPAR, statistical significance was observed for all analysed outcomes and the figure presents how it appropriately stratifies patients according to the three different cut-off points used. Finally, the CURB-65 score exhibits the most notable survival curve for in-hospital mortality; however, for ICU admission, it is remarkable that the curve for patients with intermediate scores is practically indistinguishable from the curve for low-risk patients.

### 3.4. Assessment of the Utility of suPAR Combined with the CURB-65 Score

The biomarker demonstrating superior performance (between lymphocytes and suPAR) was chosen to explore whether its combination with the CURB-65 score could enhance patient stratification. The Net Reclassification Index (NRI) was calculated to assess the improvement in risk reclassification when adding the selected biomarker to the CURB-65 score.

Table 2 presents a contingency table showing the different combinations of the risk values obtained using the CURB-65 score and those obtained for the suPAR biomarker in order to perform the Net Reclassification Improvement (NRI) analysis. Note that no patient with low suPAR developed in-hospital mortality or ICU admission outcomes.

Table 3 shows the results of the NRI analysis for the different clinical outcomes considered in this study and the combined model of the CURB-65 score and the biomarker suPAR. In-hospital mortality showed the most significant results in the reclassification of individuals with the event (NRI for events = 1.33), despite a tendency to increase false positives (NRI for non-events = −0.43). Overall, the combination results in a significant enhancement of patient risk classification (total NRI = 0.90).

On the other hand, regarding the outcome of ICU admission, there was also an improvement in the reclassification of patient risk when using the combined model of the score and biomarker (total NRI = 0.25).

Finally, statistically significant findings were also observed for the composite outcome of in-hospital mortality and/or ICU admission, although the magnitude of the association was comparatively lower (total NRI = 0.14).

## 4. Discussion

The most significant results of our study are as follows: (1) the suPAR biomarker showed high diagnostic value both for ICU admission and in-hospital mortality; (2) no patient with low suPAR (<4 ng/mL) experienced the outcomes studied; (3) the addition of the suPAR biomarker to the CURB-65 score substantially improved patient risk stratification.

The search for biomarkers capable of predicting the severity of COVID-19 infection has been increasing since the beginning of the pandemic. The evaluation of new biomarkers, such as soluble urokinase-type plasminogen activator receptor (suPAR), has acquired importance. Different studies available in the literature have explored the role of the suPAR biomarker in COVID-19 disease, finding a correlation between suPAR levels and the severity of the infection and suggesting possible cut-off points for patient risk stratification [19,26]. Nevertheless, there is no literature available that provides evidence regarding the potential of using current severity assessment scales in patients with pneumonia, such as CURB-65, together with suPAR.

Our study confirms that the most severely ill patients, those who required admission to the ICU or died during their hospital stay, had higher suPAR values than those who did not.

At the same time, our study demonstrates the good prognostic significance of suPAR for both outcomes studied (ICU admission and in-hospital mortality), which is higher than that observed for lymphocytes. As compared to the validated CURB-65 score, suPAR demonstrated better discriminatory ability for ICU admission and similar ability for in-hospital mortality.

In addition, our study found that none of the patients with low suPAR levels (<4 ng/mL) developed the considered outcomes. In the range of 4–6 ng/mL, the percentage of patients exhibiting the studied outcomes was low, suggesting a potential grey zone. Notably, with suPAR levels > 6 ng/mL, the percentage of patients presenting with ICU admission or in-hospital mortality increased substantially, highlighting the clinical relevance of higher values. These findings are consistent with the published literature, which proposes a cutoff value of 4 ng/mL as a criterion supporting patient discharge and levels above 6 ng/mL as indicator of severity [18,19]. Due to the nonspecific nature of the suPAR biomarker, which may be elevated in various pathologies, its primary utility in patients with COVID-19 pneumonia lies in its prognostic role rather than diagnostic role, effectively stratifying patients into survivors, non-survivors, or those requiring ICU admission.

Finally, the addition of the suPAR biomarker to the CURB-65 score demonstrated a considerable improvement in the reassessment of patient risk. For in-hospital mortality, the total NRI score was significantly elevated, indicating a major improvement in the reclassification of patients, either by improving the identification of those at high risk of death or by avoiding the misclassification of low-risk patients. The NRI values obtained for this outcome indicate that the use of the score alongside the biomarker correctly reclassifies 90% of patients compared to using the CURB-65 score alone. This implies that the combined model (CURB-65 score plus suPAR) has high clinical value, as it could potentially improve decision making in the management of patients with pneumonia, optimizing interventions such as hospitalisation, intensive treatment, or early discharge. Regarding ICU admission, a significant improvement in patient risk classification was observed, although the degree of reclassification, as reflected by NRI values, was more moderate than for the previous outcome. Similarly, the combination of the clinical score with the suPAR biomarker resulted in a moderate improvement in risk stratification for the composite outcome of in-hospital mortality and/or ICU admission.

Some limitations of our study must be acknowledged: (1) Our study lacks a cohort of healthy controls for biomarker level comparisons. However, for lymphocyte count, we applied the cut-off established in previous studies on patients with community-acquired pneumonia [24,25], whereas for suPAR, we used the threshold defined in various studies [18,19]. (2) Although this study includes a considerable number of patients, the sample size remains limited in terms of mortality and certain complications. (3) COVID-19 patients were enrolled at the onset of the pandemic; therefore, these findings may not be fully generalisable to patients from subsequent waves, given the impact of vaccination status, the emergence of SARS-CoV-2 variants, and changes in treatment protocols. (4) Slight increases in suPAR levels have been described in the literature depending on different comorbidities, such as diabetes, hypertension, or smoking [27]. To address this limitation, given that the greatest increase has been reported in smokers, the influence of smoking status was analysed in our cohort and no significant differences were obtained for suPAR levels. As other comorbidities were not assessed and baseline values for the patients were unavailable, slight interference of these factors on suPAR levels cannot be excluded.

## 5. Conclusions

Our findings highlight the clinical utility of the suPAR biomarker and its value in combination with severity assessment scores for pneumonia. Specifically, integrating suPAR with the CURB-65 score enables more precise risk reclassification of patients with COVID-19 pneumonia. This combined approach holds significant clinical relevance, as it may improve decision making in pneumonia management, optimizing key interventions such as hospitalisation, intensive treatment, or early discharge.

## Figures and Tables

**Figure 1 biomedicines-13-00896-f001:**
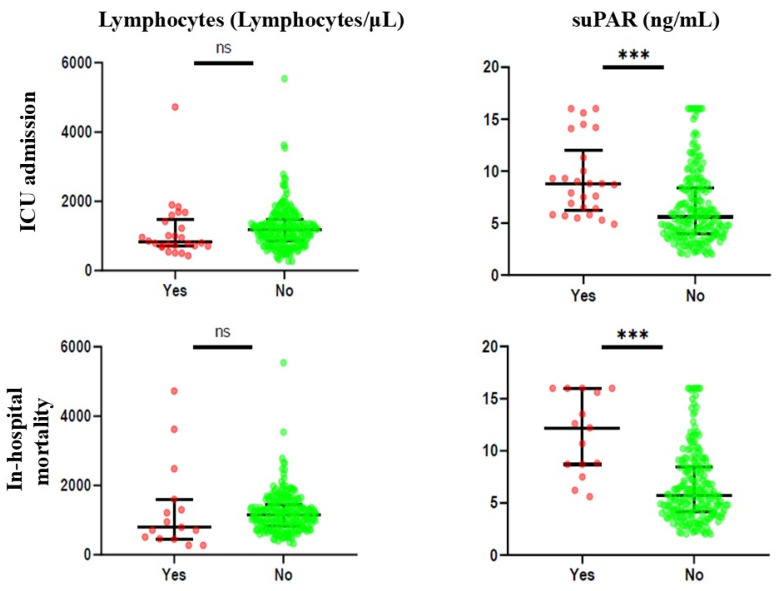
Differences in biomarker levels between patients with the presence or absence of the outcome considered. ns: not statistically significant; ***: *p* < 0.001.

**Figure 2 biomedicines-13-00896-f002:**
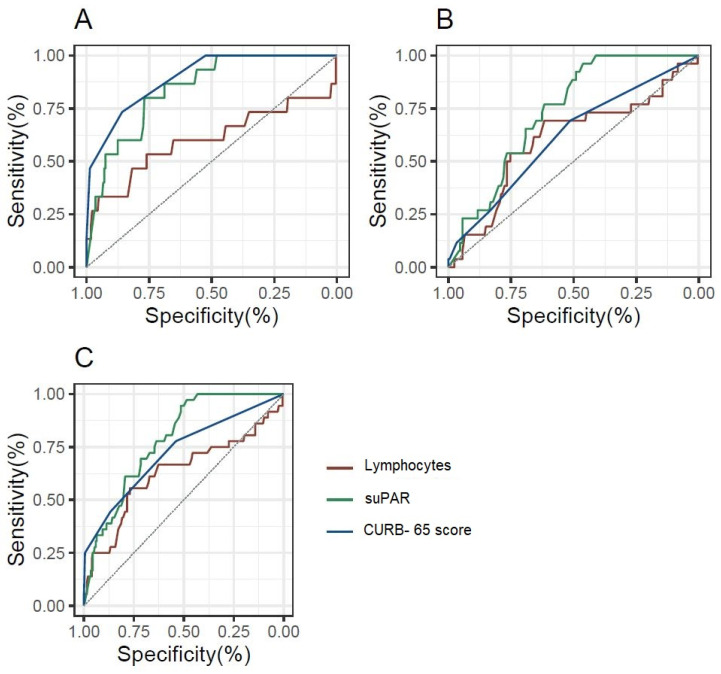
ROC curves for the diagnostic performance of lymphocytes, suPAR, and the CURB-65 score, depending on the outcome considered. (**A**): In-hospital mortality; (**B**): ICU admission; (**C**): In-hospital mortality and/or ICU admission.

**Figure 3 biomedicines-13-00896-f003:**
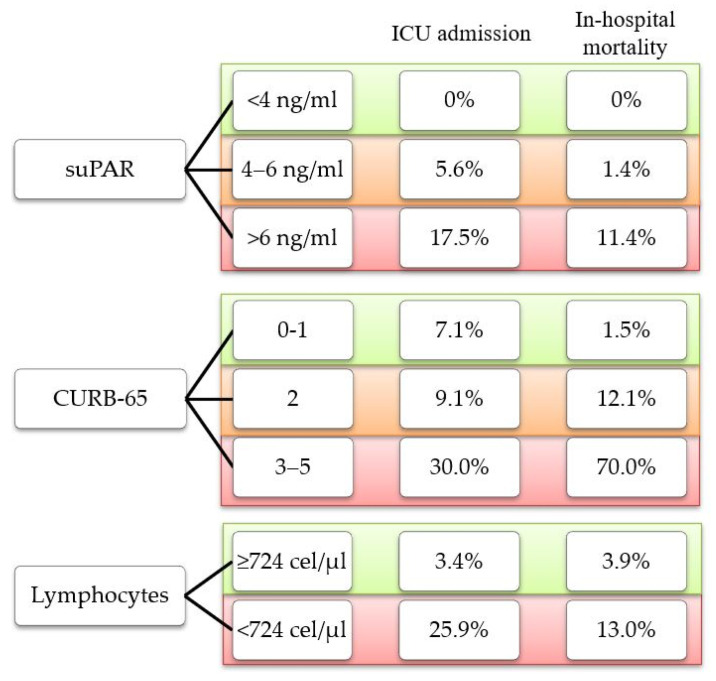
Percentage of outcome observed according to risk classification for each parameter.

**Figure 4 biomedicines-13-00896-f004:**
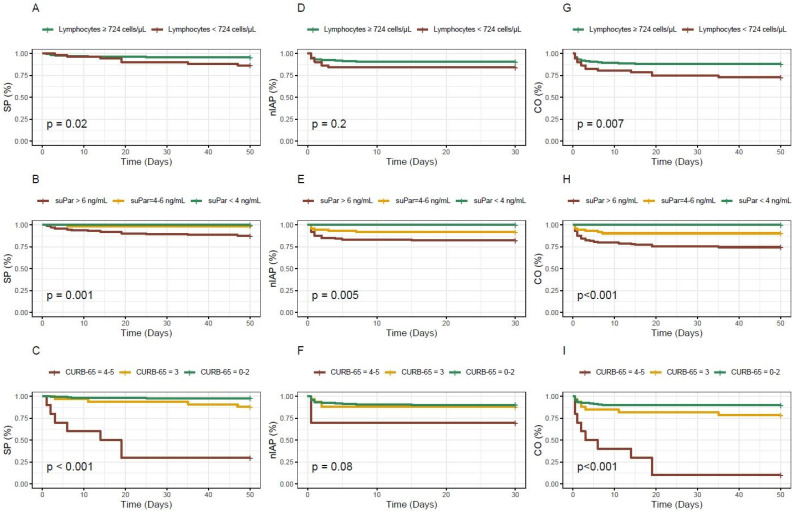
Kaplan–Meier survival curves for the different biomarkers and the CURB-65 score based on the outcome considered. (**A**–**C**): in-hospital mortality; (**D**–**F**): ICU admission; (**G**–**I**): in-hospital mortality and/or ICU admission; SP: survival probability; nIAP: non ICU admission probability; CO: combined outcome probability.

**Table 1 biomedicines-13-00896-t001:** Baseline characteristics, comorbidities, and severity.

	COVID-19 Patients with Biomarkers (*n* = 240)
Age, years, median (IQR)	55 (44, 65)
Male sex, no. (%)	125 (52.0)
Current or former smokers, no. (%)	58 (24.2)
Coexisting conditions, no. (%)	
HBP	72 (30.0)
Diabetes	40 (16.7)
Dyslipidaemia	56 (23.3)
Overweight *	113 (47.1)
COPD	10 (4.2)
Asthma	12 (5.0)
Chronic heart disease	19 (7.9)
Chronic renal disease	11 (4.6)
Neurological disease	8 (3.3)
Severity	
Days of admission, median (IQR)	8 (6, 12)
SpO_2_/FiO_2_ at admission, median (IQR)	452.0 (433.3, 457.1)
CURB-65 score, no. (%)	
0–1 (Low risk)	197 (82.1)
2 (Intermediate risk)	33 (13.8)
3–5 (High risk)	10 (4.2)
Days of symptoms prior admission, median (IQR)	7 (4, 9)
Clinical outcomes, no. (%)	
ICU admission	26 (10.8)
In-hospital Mortality	15 (6.3)
In-hospital mortality and/or ICU admission	36 (15.0)

IQR: interquartile range; HBP: high blood pressure; COPD: chronic obstructive pulmonary disease; SpO_2_/FiO_2_: peripheral blood oxygen saturation/fraction of inspired oxygen; CURB-65: clinical scoring system used to assess the severity of pneumonia; ICU: intensive care unit. * Body mass index ≥ 25.

**Table 2 biomedicines-13-00896-t002:** Contingency table for the combined CURB-65 score and suPAR model for the different clinical outcomes.

Clinical Outcome	CURB-65 Score	suPAR	Total Patients	Patients with the Outcome (%)
In-hospital mortality	high	high	10	7 (70)
	intermediate	high	23	4 (17)
	intermediate	intermediate	5	0
	intermediate	low	5	0
	low	high	81	3 (4)
	low	intermediate	68	1 (1)
	low	low	48	0
ICU admission	high	high	10	3 (30)
	intermediate	high	23	3 (13)
	intermediate	intermediate	5	1 (20)
	intermediate	low	5	0
	low	high	81	14 (17)
	low	intermediate	68	5 (7)
	low	low	48	0
Composite outcome of in-hospital mortality and/or ICU admission	high	high	10	9 (90)
	intermediate	high	23	6 (26)
	intermediate	intermediate	5	1 (20)
	intermediate	low	5	0
	low	high	81	14 (17)
	low	intermediate	68	6 (9)
	low	low	48	0

**Table 3 biomedicines-13-00896-t003:** Results of the Net Reclassification Improvement (NRI) analysis for the combined CURB-65 and suPAR model.

Clinical Outcome	NRI for Events	NRI for Non-Events ^1^	Total NRI
In-hospital mortality	1.33	−0.43	0.90
ICU admission	0.65	−0.41	0.25
Composite outcome of in-hospital mortality and/or need for ICU admission	0.56	−0.41	0.14

^1^ Negative NRI values for non-events suggest misclassification, where true non-events were reclassified into higher-risk categories, suggesting reduced specificity.

## Data Availability

The datasets used and/or analysed during the current study are available from the corresponding author upon reasonable request.

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
