# Peer review of "Evaluation of suPAR as a Key Prognostic Biomarker in Patients with SARS-CoV-2 Pneumonia"

_biomedicines, 2025, doi:10.3390/biomedicines13040896_

Round 1
Reviewer 1 Report
Comments and Suggestions for Authors
The manuscript entitled “Evaluation of suPAR as a Key Prognostic Biomarker in Patients 2 with SARS-CoV-2 Pneumonia” has some medical value however, it contains some gaps.
The article is generally well written and easy to understand. There are a few comments.
Given that suPAR is a non-specific marker of severity and aggressiveness of many diseases and is associated with morbidity and mortality in a number of acute and chronic diseases, one of the major gaps in the paper is the lack of correlation of serum SuPAR with the presence or absence of individual clinical or virological characteristics in the studied SARS Cov-2 patients. This does not allow us to understand the relationship of key clinical characteristics with SuPAR levels.
The article contains almost no data on comorbidities that may also affect SuPAR levels.
Given the low specificity of the SuPAR level, are misdiagnoses possible? In other words, in patients with non-coronavirus pathology (but positive for the virus - in SARS Sov-2 carriers) and with a high SuPAR level, for example, with systemic inflammatory response syndrome, are diagnostic and monitoring errors possible? It seems appropriate to discuss this issue in the discussion section.
Minor points
What do the authors mean by the term "inclusion criteria"? Are these their original criteria or are they taken from the literature. In any case, this term requires additional explanation.
The same remark applies to the term "exclusion criteria". What do the authors mean by the term immunosuppression?
The provision "data collection protocol" needs to be clarified.
Author Response
Dear Reviewer,
Thank you very much for the insightful comments provided on our manuscript. Your suggestions have significantly improved the quality of the work. I will now address each point raised in your review and detail the changes that have been made accordingly.
Comments 1: Given that suPAR is a non-specific marker of severity and aggressiveness of many diseases and is associated with morbidity and mortality in a number of acute and chronic diseases, one of the major gaps in the paper is the lack of correlation of serum SuPAR with the presence or absence of individual clinical or virological characteristics in the studied SARS Cov-2 patients. This does not allow us to understand the relationship of key clinical characteristics with SuPAR levels.
Response 1: Thank you very much. As you have correctly pointed out, the suPAR biomarker is nonspecific, which is why it is not suitable for diagnostic purposes. Nevertheless, its primary clinical value lies in prognosis, as it enables effective risk stratification by identifying non-survivors and patients requiring ICU admission versus those who do not.
We have added a comment in section 4. Discussion of the manuscript regarding the primary use of suPAR as a biomarker and its limitation as a diagnostic biomarker.
Comments 2: The article contains almost no data on comorbidities that may also affect SuPAR levels.
Response 2: Thank you for your feedback. In response, we have tested the influence of smoking status (smoker/non-smoker) in our cohort, as the literature reports the highest increases in suPAR levels in smokers. However, no statistically significant differences were found, despite smokers having a slightly higher median suPAR level compared to non-smokers. These results are not included in the manuscript because of similar data but we have added an explanation in discussion section (4. Discussion - Limitations acknowledgement). As other comorbidities were not assessed and baseline values for the patients were unavailable, a slight interference of these factors on suPAR values cannot be excluded. Nonetheless, as values above 6 ng/mL are consistently associated with critical illness, these minor variations can be considered non-impactful and do not compromise its prognostic utility in acute COVID-19 pneumonia.
Smoker status YES/NO median value and p (Mann-Whitney)
YES 6.7 (4.8-10.8)
NO 5.7 (4.1-8.4)
p=0.061
Comments 3: Given the low specificity of the SuPAR level, are misdiagnoses possible? In other words, in patients with non-coronavirus pathology (but positive for the virus - in SARS Sov-2 carriers) and with a high SuPAR level, for example, with systemic inflammatory response syndrome, are diagnostic and monitoring errors possible? It seems appropriate to discuss this issue in the discussion section.
Response 3: Thank you for your clarification. As all patients included in the study had COVID-19 pneumonia (defined by newly developed infiltrates on chest X-ray alongside a positive SARS-CoV-2 test), we consider the systemic inflammatory response to be associated with the pneumonic process. Furthermore, we have specified in the manuscript that the primary utility of suPAR as a biomarker is prognostic rather than diagnostic (section 4. Discussion).
Comments 4: What do the authors mean by the term "inclusion criteria"? Are these their original criteria or are they taken from the literature. In any case, this term requires additional explanation.
Response 4: Thank you very much for your observations. We have included an explanation in section 2.1 Study design and participation of the manuscript regarding the inclusion and exclusion criteria considered in the study.
Comments 5: The same remark applies to the term "exclusion criteria". What do the authors mean by the term immunosuppression?
Response 5: Thank you very much for your appreciation. As in the previous point, we have included the corresponding explanation regarding the inclusion and exclusion criteria in section 2.1 Study design and participation of the manuscript, as well as the criteria for considering immunosuppression.
Comments 6: The provision "data collection protocol" needs to be clarified.
Response 6: The research team responsible for the study adheres to a standardised data collection protocol to ensure the systematic and anonymised recording of all variables under investigation, aiming to minimise data loss as much as possible. This protocol includes the collection of initial data to determine whether inclusion criteria are met, as well as information on medical history, prior treatments, comorbidities, initial severity, diagnosis, clinical data at admission and during hospitalisation, physical examination findings, laboratory analyses, radiological assessments, in-hospital treatments, and patient outcomes.

Reviewer 2 Report
Comments and Suggestions for Authors
The manuscript entitled Evaluation of suPAR as a Key Prognostic Biomarker in Patients 2 with SARS-CoV-2 Pneumonia is an interesting piece of work that highlights the significance of biomarkers associated with the certain health issues. However, the quality of the manuscript can be improved by addressing the following queries:
- Please briefly define the term “CURB-65 score” before using it.
- The statement “Limitation acknowledgement …” (Lines 284-291) would be better placed in the Materials and Methods section for clarity and the same lines (284-291) can be omitted.
- In Table 3, please add a footnote to explain the relevance of Negative Risk Index (NRI) for non-events in terms of their negative values.
- In the Results sections (3.2, 3.3 and 3.4), please provide information on the relevance of experimentation performance before proceeding to the results analysis.
- As shown in Figure 1, which compares ICU admission vs lymphocyte concentration (cells/μL), please correct the caption to read 'lymphocytes per microliter.
- Finally, the manuscript's literature should be updated to include research from 2025.
Author Response
Dear Reviewer,
Thank you for your valuable feedback on our manuscript. The application of your suggestions has undoubtedly enhanced its quality. I will now address each of the points raised in your review and outline the corresponding changes made.
Comments 1: Please briefly define the term “CURB-65 score” before using it.
Response 1: Thank you very much for your clarification. We have added in section 2.1 Study design and participation of the manuscript an explanation of what means CURB-65 and its utility.
Comments 2: The statement “Limitation acknowledgement …” (Lines 284-291) would be better placed in the Materials and Methods section for clarity and the same lines (284-291) can be omitted.
Response 2: Thank you for your valuable appreciation. Following the structure of recently published research articles by the journal, we have decided to place the "Limitations Acknowledgment [..]" at the end of the Discussion section
Comments 3: In Table 3, please add a footnote to explain the relevance of Negative Risk Index (NRI) for non-events in terms of their negative values.
Response 3: Thank you for your insightful remark. It has been added a footnote in table 3, explaining the negative values for NRI non-events.
Comments 4: In the Results sections (3.2, 3.3 and 3.4), please provide information on the relevance of experimentation performance before proceeding to the results analysis.
Response 4: Thank you for the feedback. An explanation regarding experimentation performance has been added before discussing the results in sections 3.2, 3.3, and 3.4 to enhance clarity.
Comments 5: As shown in Figure 1, which compares ICU admission vs lymphocyte concentration (cells/μL), please correct the caption to read 'lymphocytes per microliter.
Response 5: Thank you for your contribution. We have revised Figure 1, replacing (cells/µL) with (lymphocytes/µL).
Comments 6: Finally, the manuscript's literature should be updated to include research from 2025.
Response 6: Thank you for your contribution. A recently published reference relevant to our study has been incorporated. [21]

Reviewer 3 Report
Comments and Suggestions for Authors
This manuscript evaluates the prognostic value of suPAR in COVID-19 pneumonia compared to leukocyte count and CURB-65, and explores its additive utility in risk stratification. The data were collected during the early pandemic, however, viral variants, vaccination status, and treatment protocols have since evolved, which may limit the generalizability of results. The study does not compare suPAR to other established COVID-19 biomarkers ,which limits the reader’s ability to contextualize suPAR’s relative value.
Author Response
Dear Reviewer,
Thank you for your constructive feedback on our manuscript. Your comments have greatly contributed to the improvement of the quality of our work. I will now address each point raised in your review and provide a summary of the modifications made.
Comments 1: This manuscript evaluates the prognostic value of suPAR in COVID-19 pneumonia compared to leukocyte count and CURB-65, and explores its additive utility in risk stratification. The data were collected during the early pandemic, however, viral variants, vaccination status, and treatment protocols have since evolved, which may limit the generalizability of results. The study does not compare suPAR to other established COVID-19 biomarkers, which limits the reader’s ability to contextualize suPAR’s relative value.
Response 1: Thank you for your valuable contribution. Point (3) in the Limitations Acknowledgment section has been revised.
Regarding the use of alternative biomarkers, we compared suPAR with lymphocytes, as lymphopenia emerged as a significant prognostic biomarker in COVID-19 and was widely utilised during the pandemic. However, in our cohort, lymphocyte levels did not demonstrate statistically significant results. Similarly, we incorporated the CURB-65 prognostic score due to its extensive validation in community-acquired pneumonia and its widespread application during the pandemic for initial patient stratification. Our study focused on assessing the utility of suPAR alone and evaluating whether its combination with the CURB-65 score enhances patient stratification.
